# Bridging the Gap between Semantic Correspondence and Robust Visual Representation

## Abstract

Predicting cross-image semantic correspondence among various instances within the same category is a fundamental but challenging task in computer vision. Models are supposed to characterize both high-level semantic features and low-level texture information to accurately finds the correspondence between pixels. The quality of features directly affects the matching results. Recently, pre-trained models with self-supervised training methods have demonstrated promising performance in representation learning and can serve as a strong backbone to provide robust visual features. However, existing methods have been found to poorly adapt to such features. Their complex designs of the matching module do not yield significant performance boost due to the disruption of the original representation and the absence of high-resolution low-level information. In this work, we introduce a simple yet effective framework named ViTSC to unlock the substantial potential of self-supervised vision transformers for semantic correspondence. We introduce three key components: a cross-perception module to align semantic features of the same part from different images while preserving the original representation as much as possible, an auxiliary loss to eliminate ambiguity from semantically similar objects, and a low-level correlation-guided upsampler to generate high-resolution flow maps for precise localization. ViTSC shows reliable semantic correspondence performance, surpassing previous state-of-the-art methods on all three standard benchmarks SPair-71k, PF-PASCAL and PF-WILLOW.

## 1 Introduction

Semantic correspondence prediction is a fundamental task in computer vision, holding significant implications for tasks such as image classification (Zhang et al., 2020; Afrasiyabi et al., 2022), few-shot segmentation (Min et al., 2021; Liu et al., 2023), video object segmentation (Hu et al., 2018; Seong et al., 2021), object tracking (Zhu et al., 2016; Nebehay & Pflugfelder, 2014), and beyond. The objective of semantic correspondence is to establish correspondences between two images. Particularly, semantic correspondence emphasizes the matching of distinct objects belonging to the same category. In contrast, other matching tasks, such as optical flow, typically focus on matching the same object across different frames. Therefore, except for low-level features, semantic correspondence is a task that demands high-level semantic information, such as category information. Historically, due to the absence of a universally robust visual backbone providing holistic high-level semantic information, semantic correspondence remains an unresolved challenge.

A typical model for semantic correspondence usually consists of a backbone and a matching module. The backbone extracts features from the images. Then the matching module computes a correlation matrix using these features. Through a series of operations, the features and correlation matrix are enhanced, ultimately uncovering the matching results. Previous works (Seung Wook Kim, 2022; Cho et al., 2021; 2022; Sun et al., 2023; Min et al., 2020; Min & Cho, 2021) primarily focus on the design of the matching module, while neglecting the fact that the quality of the extracted features directly impacts the matching results. Most existing matching networks rely on convolutional neural networks (CNN) (He et al., 2016; Simonyan & Zisserman, 2014) pre-trained on the image classification task using the ImageNet dataset as the backbone, and the quality of the extracted features is insufficient for generating satisfactory matching results directly. Therefore, the matching modules in previous works are designed to be complex, resulting in high computational overhead. In recent times, with the emergence of Vision Transformer (Dosovitskiy et al., 2021) (ViT) and various self-

supervised pre-training methods (He et al., 2022; Radford et al., 2021; Zhou et al., 2021; Caron et al., 2021; Oquab et al., 2023; Bao et al., 2022; Xie et al., 2022; Fang et al., 2023), more robust features can be obtained. Different pre-trained models possess distinct characteristics, e.g., MAE (He et al., 2022) has strong inpainting ability, while CLIP (Radford et al., 2021) shows significant zero-shot classification performance. These capabilities emerging from self-supervised pre-training make it feasible to replace the previously widely adopted CNN with a more powerful pre-trained backbone.

In this work, we conduct comprehensive experiments to explore the impact of different visual pre-trained models and matching modules. We find that the pre-trained models utilizing masked modeling perform better on semantic correspondence than the models pre-trained via text-image contrastive learning. We believe that masked image modeling forces the model to learn distinguishable local representations. Subsequently, using the pre-trained model with the best initial features as the backbone, we test whether various matching modules can adapt to the new robust features in a manner similar to adapting to CNN features. However, we observe that the complex matching modules proposed in previous works are no longer applicable due to their excessive disruption of the original representation and the lack of high-resolution low-level features. In comparison to a very simple matching module, the performance improvement from more complex matching modules is limited, or even negative in some cases, despite the increased computational complexity.

In order to design a matching module that is more suitable for the more powerful features, we aim for the matching module to address the following issues: 1) aligning features of the same parts across different objects, 2) distinguishing features between highly similar parts, and 3) accurately localizing parts with specific semantics. To achieve this, we 1) design a cross-perception module to enable two images to have mutual perception, thereby making the features of the same parts in different objects more similar, 2) introduce an auxiliary loss to differentiate highly similar parts, 3) design a high-resolution low-level correlation-guided upsampling module to achieve more precise localization.

The core contributions of our work can be summarized as follows:

- To adapt to the recently emerged robust visual representation, we design a cross-perception module for feature enhancement, introduce an auxiliary loss to discriminate similar objects, and design a high-resolution low-level correlation-guided upsampling module for precise localization. They form a matching model utilizing the power of the backbones effectively.

- We construct a simple baseline to evaluate the transferability of pre-trained backbones. By conducting systematic experiments on various backbones and matching modules, we reveal that the quality of the features from pre-trained models has a significant impact on the matching results and the complex matching modules proposed in previous works are no longer applicable to the powerful features available today.

- Experimental results demonstrate that our model surpasses the state-of-the-art methods in all three popular semantic correspondence benchmarks SPair-71k, PF-PASCAL and PF-WILLOW. Specifically, our method gains 3.5 PCK@0.10 improvement on SPair-71k.

## 2 RELATED WORK

**Semantic Correspondence.** Existing methods typically begin by utilizing a backbone network to extract features from the images, followed by a matching module that predicts the correspondence between the images based on backbone features. Recently, numerous matching methods have been proposed, which can be categorized into two classes: CNN-based methods and Transformer-based methods. CNN-based often utilize convolution to extract. Some of these methods (Rocco et al., 2018; Li et al., 2020; Salehi & Balasubramanian, 2023; Hong et al., 2022b;c) utilize 4D convolution to promote neighbourhood consensus within the correlation matrix. CHMNet (Min & Cho, 2021) employs a learnable geometric matching algorithm in combination with 6D convolution to establish visual correspondence. DHPF (Min et al., 2020) employs an adaptive architecture to dynamically exploit multi-scale features during inference. NeMF (Hong et al., 2022c) represents the correlation matrix in an arbitrary resolution with an implicit neural field. In contrast, Transformer-based methods typically employ attention-based approaches to establish correspondences between images. Some of these approaches concentrate on using Transformer to enhance the features (Sun et al., 2023; Hong et al., 2022a) or refine the correlation matrix (Cho et al., 2021; 2022; Hong et al.,

2022a). TransforMatcher (Seung Wook Kim, 2022) proposes match-to-match attention to refine the initial matches. ACTR (Sun et al., 2023) uses Transformer for matching flow super-resolution. Recent studies (Sun et al., 2023; Li et al., 2023) have leveraged ViTs as their backbone due to their powerful expressive ability, resulting in significant performance boosts. The use of Transformers enables these models to comprehend global semantics more robustly by capturing long-range relations.

**Self-supervised Pre-training.** Self-supervised learning techniques, such as contrastive learning (He et al., 2020; Chen et al., 2020; Caron et al., 2020; Grill et al., 2020; Caron et al., 2021; Oquab et al., 2023) and masked image modeling (Bao et al., 2022; Zhou et al., 2021; He et al., 2022; Xie et al., 2022; Fang et al., 2023), have become common approaches for pre-training both CNNs and ViTs in recent years. Self-supervised pre-trained models can achieve impressive performance with or without fine-tuning on various downstream tasks. Furthermore, certain studies (Amir et al., 2021) have discovered the emergence of semantic correspondence capabilities in pre-trained models. Some recent work is investigating the utilization of self-supervised trained generative models for perception tasks. In semantic matching tasks, DIFT (Tang et al., 2023) utilizes the U-Net in a Stable Diffusion (Rombach et al., 2022) model as a feature extractor without fine-tuning. SD-DINO (Zhang et al., 2023) combines the features of Stable Diffusion and DINOv2 to complement each other. GeoAware-SC (Zhang et al., 2024) further improved the performance of SD+DINO through training. Diffusion Hyperfeatures (Luo et al., 2023) aim to simultaneously leverage multi-scale and multi-timestep Stable Diffusion features and train a aggregation network to obtain enhanced semantic features. These methods employ pretrained models directly for predictions, demonstrating remarkable zero-shot transferability capabilities.

# 3 AN EMPIRICAL STUDY ON BACKBONES AND MATCHING MODULES

In this section, we initially establish a simple baseline to compare the performance of various pre-trained backbones (Zhou et al., 2021; He et al., 2022; Radford et al., 2021; Oquab et al., 2023) on semantic correspondence, then assess the compatibility between the recently emerged powerful backbones and various matching modules proposed in previous works.

## 3.1 A SIMPLE MATCHING BASELINE

A network can perform the following typical steps to accomplish matching tasks. Given a source image $\mathbf{I}^{\mathrm{A}} \in \mathbb{R}^{H \times W \times 3}$ annotated with the location of source keypoints $\mathbf{K}^{\mathrm{A}}$ and a target image $\mathbf{I}^{\mathrm{B}} \in \mathbb{R}^{H \times W \times 3}$, the network is designed to predict the target keypoint location $\hat{\mathbf{K}}^{\mathrm{B}}$ corresponding to the source keypoints $\mathbf{K}^{\mathrm{A}}$ for the target image $\mathbf{I}^{\mathrm{B}}$. $\mathbf{K}^{\mathrm{A}}$ can be dense or sparse as the task requires.

Specifically, a vision backbone network first extracts feature maps $\mathbf{F}^{\mathrm{A}}, \mathbf{F}^{\mathrm{B}} \in \mathbb{R}^{(\frac{H}{S} \times \frac{W}{S}) \times C}$ from two input images $\mathbf{I}^{\mathrm{A}}$ and $\mathbf{I}^{\mathrm{B}}$, respectively; $C$ denotes the output dimension of the vision backbone; $S$ denotes the stride of the feature map. Here, we uniformly upsample all feature maps $\mathbf{F}^{\mathrm{A}}, \mathbf{F}^{\mathrm{B}}$ to a stride of $S = 8$. Next, the 2D correlation matrix $\mathbf{C} \in \mathbb{R}^{(h \times w) \times (h \times w)}$, where $(h, w) = (\frac{H}{8}, \frac{W}{8})$, can be calculated by cosine similarity as follows:

$$\mathbf{C} = \mathrm{CosineSim}(\mathbf{F}^{\mathrm{A}}, \mathbf{F}^{\mathrm{B}}) = \frac{\mathbf{F}^{\mathrm{A}} \cdot \mathbf{F}^{\mathrm{B}^{T}}}{||\mathbf{F}^{\mathrm{A}}|| \cdot ||\mathbf{F}^{\mathrm{B}}||}, \tag{1}$$

$\mathbf{C}$ contains the correlation scores between all pixels in the source feature map $\mathbf{F}^{\mathrm{A}}$ and all pixels in the target feature map $\mathbf{F}^{\mathrm{B}}$.

The prediction keypoints can be extracted from the correlation matrix $\mathbf{C}$ using the following soft-argmax procedure. A two-dimensional Gaussian kernel is firstly applied to the $\mathbf{C}$ to suppress non-maximum local maxima like in (Lee et al., 2019). Then $\mathbf{C}$ is normalized by softmax operation along the second dimension to get the correlation distribution $\mathbf{P} \in \mathbb{R}^{(h \times w) \times (h \times w)}$:

$$\mathbf{P} = \mathrm{Softmax}(\frac{\mathbf{C}}{\mathcal{T}}), \tag{2}$$

where $\mathcal{T}$ is a temperature coefficient introduced to prevent excessive smoothing of the correlation matrix. Inspired by (Li et al., 2023), the temperature is configured as a learnable parameter which is

| Backbone | PCK@$\alpha_{bbox}$* | | | PCK@$\alpha_{bbox}$ | | |
|---|---|---|---|---|---|---|
| | 0.01 | 0.05 | 0.10 | 0.01 | 0.05 | 0.10 |
| iBOT | 0.7 | 14.3 | 32.5 | 4.3 | 40.2 | 59.9 |
| MAE | 0.1 | 2.6 | 8.1 | 4.8 | 43.2 | 63.4 |
| CLIP | 0.2 | 4.9 | 13.4 | 3.9 | 39.8 | 58.9 |
| DINOv2 | **1.3** | **20.2** | **40.6** | **7.0** | **58.6** | **78.2** |

Table 1: Evaluation results of the simple baseline with different backbones on SPair-71k. * indicates all parameters of the model have not been fine-tuned on the dataset.

| Matching Module | Backbone | PCK@$\alpha_{bbox}$ | | |
|---|---|---|---|---|
| | | 0.01 | 0.05 | 0.10 |
| Baseline | DINOv2 | **7.0** | **58.6** | 78.2 |
| TransforM. | DINOv2 | 6.1 | 56.1 | **78.3** |
| CATs++ | DINOv2 | 4.1 | 49.3 | 74.9 |
| ACTR | DINOv2 | 4.2 | 55.6 | 74.9 |

Table 2: Evaluation results of various methods using DINOv2-B as the backbone. All models are fine-tuned on SPair-71k.

initialized as 0.03 and can be optimized during the training procedure. Then dense correspondence keypoint predictions $\hat{\mathbf{G}} \in \mathbb{R}^{(h \times w) \times 2}$ are calculated by employing matrix multiplication on $\mathbf{P}$ and the 2D coordinate grid $\mathbf{G} \in \mathbb{R}^{(h \times w) \times 2}$:

$$\hat{\mathbf{G}} = \mathbf{P} \cdot \mathbf{G}, \tag{3}$$

and we can obtain the matching flow map $\mathbf{M} \in \mathbb{R}^{(h \times w) \times 2}$ by:

$$\mathbf{M} = \hat{\mathbf{G}} - \mathbf{G} \tag{4}$$

After $\mathbf{M}'$ is generated by upsampling the flow map to the full resolution, the dense correspondence from each pixel in the source image to the target image is obtained. Sparse prediction keypoints $\hat{\mathbf{K}}^{\mathrm{B}}$ can be extracted from $\hat{\mathbf{G}}$ with index selecting. Similarly, sparse matching flow can be extracted from $\mathbf{M}'$ with index selecting.

The simplest and most intuitive approach to train this model is to supervise with L2 loss between the source keypoints and target keypoints. However, to enhance local consistency in the matching results, we employ a training strategy that involves pseudo optical flow generated from keypoint annotations as (Cho et al., 2021) does. The model is supervised using matching flow information between the source keypoints and their neighboring keypoints within a rectangular range surrounding them. We generate pseudo flow by:

$$\mathcal{F}_{\mathcal{N}(\mathbf{p})} = \mathcal{N}(\mathbf{p}) + \mathbf{q} - \mathbf{p}, \tag{5}$$

where $\mathbf{p}, \mathbf{q} \in \mathbb{R}^{1 \times K \times 2}$ are coordinates of the source keypoints and the target keypoints, and $\mathcal{N}(\mathbf{p}) \in \mathbb{R}^{l^2 \times K \times 2}$ represents coordinates of the points in the rectangular neighborhood of $\mathbf{p}$. During training, we optimize the network by minimizing the end point error (EPE) loss between the ground-truth flow and the predicted flow, defined as

$$\mathcal{L}_{\mathrm{EPE}} = \frac{1}{N} \sum \|\hat{\mathcal{F}} - \mathcal{F}\|, \tag{6}$$

where $\mathcal{F}$ and $\hat{\mathcal{F}}$ are the ground-truth and predicted flow, and $N$ is the number of supervised pixels.

## 3.2 Preliminary Experiments and Analyses

We use the above model and training process as a simple baseline. In this simple baseline, we do not conduct any feature aggregation or correlation matrix aggregation methods in the matching module, thus the crucial factor for achieving good performance lies in feature quality of the features. When the backbone is initialized with a strong pre-trained model capable of generating high-quality features, the model is more inclined to learn accurate visual correspondence.

In our preliminary experiment, we train and evaluate this simple baseline initialized with four popular self-supervised pre-trained ViTs: iBOT (Zhou et al., 2021), MAE (He et al., 2022), CLIP (Radford et al., 2021) and DINOv2 (Oquab et al., 2023) on a standard semantic correspondence benchmark SPair-71k. The results in Tab. 1 reveal significant variations in the performance of different backbones. When pre-trained models are applied directly to semantic correspondence tasks, DINOv2 and iBOT exhibit some level of matching ability. However, MAE and CLIP do not demonstrate the same level of proficiency in these tasks initially. After fine-tuning the models on SPair-71k, all pre-trained models experience a notable performance boost. Among the four options, DINOv2

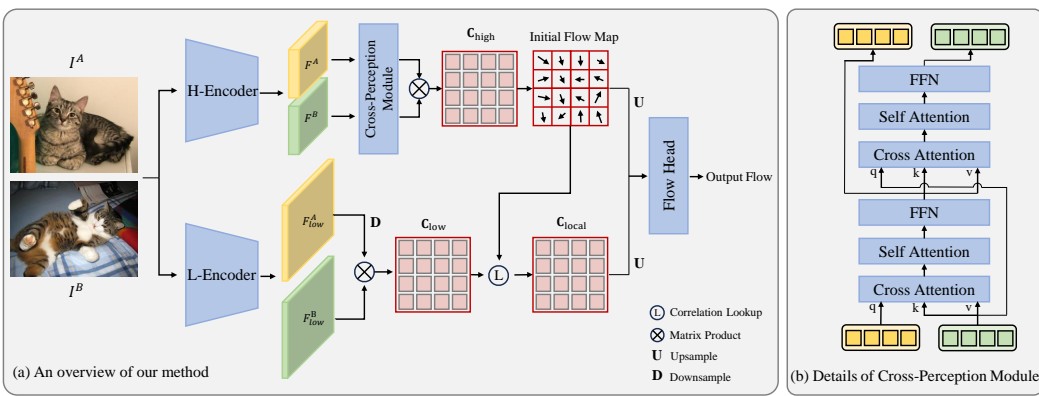

Figure 1: The framework of ViTSC. (a) ViTSC extracts features with a high-level semantic encoder, enhances the features with an cross-perception module, and upsamples the initial matching flow with local correlation generated from a low-level texture encoder. (b) The cross-perception module enables the source features and the target features have a mutual perception.

features stand out when frozen and have a higher performance upper bound when fine-tuned because DINOv2 learns all-purposed features at the patch-level and image-level using large-scale curated data. This allows DINOv2 to perform effective feature matching between image patches. Although the other three backbones may not perform as well as DINOv2 initially, they still get objective performance gains after fine-tuning. Particularly intriguing is the experiment involving MAE, wherein the model using original MAE pre-trained weights exhibits notably poor matching accuracy (lowest among the four). However, after fine-tuning, the MAE model surpasses both the models with iBOT and CLIP in terms of accuracy. The above observations provide us with two key insights: 1) It is important and necessary to fine-tune the models to bridge the gap due to different optimization objectives between pre-training and semantic correspondence. 2) Masked image modeling is more suitable for semantic correspondence than text-image contrastive learning because it forces the model to learn distinguishable local representations.

Furthermore, we employed DINOv2 as the backbone since it achieves the best performance, and tested various matching modules proposed in previous works (Seung Wook Kim, 2022; Sun et al., 2023; Cho et al., 2022) to assess their compatibility with it (see Tab. 2). All models were trained on SPair-71k under the same settings, describeb in the appendix. Results show that very limited performance improvement is obtained by replacing the simple matching module with previous state-of-the-art matching modules. Some of them even perform worse than the simple baseline although they bring more computational complexity. These methods often incorporate complicated feature or correlation matrix aggregation modules, which may adversely affect the original robust representation initially generated by the backbone. With this intuition in mind, we aim to design a simple yet effective matching module that can adapt to stronger features, preserving their powerful semantic properties while making them more suitable for the semantic correspondence task.

## 4 METHOD

### 4.1 CROSS-PERCEPTION MODULE

Given the vital importance of feature quality in visual correspondence, feature enhancement is a common approach (Sun et al., 2023; Xu et al., 2022) used to obtain features with increased semantic information. In GMFlow (Xu et al., 2022), the symmetric method treats the features of the source image and the target image in a completely equivalent manner, allowing for mutual information exchange. Conversely, in ACTR (Sun et al., 2023), the asymmetric method updates the features in the source image using the features from the target image. In our work, we propose a simple but effective interleaved attention module that significantly enhances the expression capability of features with just one single layer.

The interleaved attention module is designed to integrate the features $\mathbf{F}^A$ and $\mathbf{F}^B$. As shown in Fig. 1(b), the inputs to the interleaved attention module are the feature maps $\mathbf{F}^A$ and $\mathbf{F}^B$. These feature

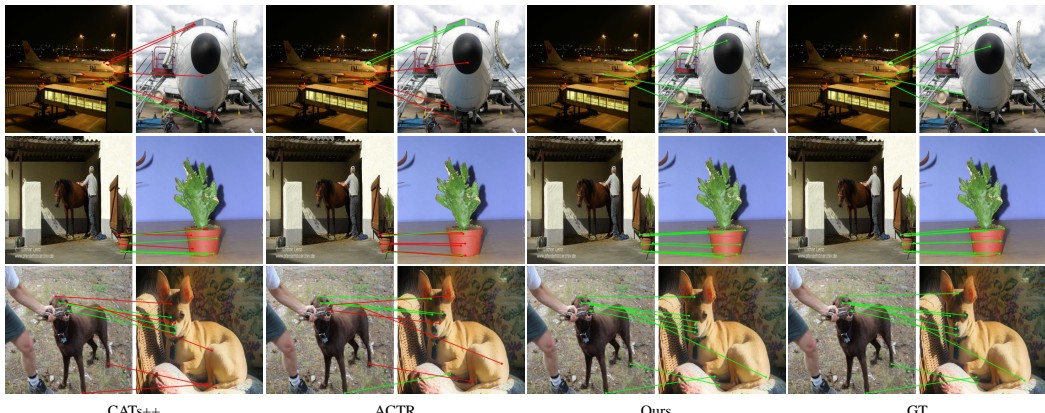

CATs++          ACTR          Ours          GT

Figure 2: The qualitative comparison. Correct matches are marked as green lines while incorrect matches are marked as red lines. Our method show better results compared with previous methods in difficult scenes.

maps sequentially pass through two attention blocks, each composed of a cross-attention layer, a self-attention layer, and a feed-forward network in sequential order. The cross-attention layer is employed to fuse features across frames, while the self-attention layer is utilized to update features within each frame. In contrast to previous methods, we find that alternately updating $\mathbf{F}^A$ and $\mathbf{F}^B$ leads to improved performance. Specifically, in the first attention block of the interleaved attention module, $\mathbf{F}^A$ serves as the query, while $\mathbf{F}^B$ acts as the key and value. Consequently, features from $\mathbf{F}^B$ are initially integrated into $\mathbf{F}^A$. Conversely, in the subsequent second attention block, $\mathbf{F}^B$ functions as the query, and $\mathbf{F}^A$ serves as the key and value. This results in the features from $\mathbf{F}^A$ being initially fused into $\mathbf{F}^B$. This process can be described as follows:

$$\mathbf{F}^{A'} = \mathrm{FFN}(\text{Self-Attn}(\text{Cross-Attn}(\mathbf{F}^A, \mathbf{F}^B))),$$
$$\mathbf{F}^{B'} = \mathrm{FFN}(\text{Self-Attn}(\text{Cross-Attn}(\mathbf{F}^B, \mathbf{F}^{A'}))). \tag{7}$$

Positional encoding and layer normalization are omitted in Eq. 7 for simplicity. Standard multi-head attention is used to implement this module. By feeding $\mathbf{F}^{A'}$ and $\mathbf{F}^{B'}$ back as inputs to another instance of this module, this module can be stacked to form multiple layers.

## 4.2 CORRELATION-GUIDED UPSAMPLER

Different from previous methods that supervise the flow map at a low resolution, we supervise the flow map at the full resolution, allowing the model to capture precise information. However, directly interpolating the flow map bilinearly presents two problems: 1) the upsampled flow map tends to be oversmoothing, potentially losing important details, and 2) the interpolated flow may not be reasonable at the edge of objects. To address these challenges, we propose a low-level correlation guided correlation-guided upsampler that upsamples the flow map with fine-grained correlations.

This module utilizes two sets of features: high-level semantic features $\mathbf{F}_{\mathrm{high}} = [\mathbf{F}^A, \mathbf{F}^B]$ extracted by H-Encoder and low-level texture features $\mathbf{F}_{\mathrm{low}}$ extracted by L-Encoder. In this context, the H-Encoder refers to the pretrained backbone utilized for extracting feature maps $\mathbf{F}^A$, $\mathbf{F}^B$, with typically lower output resolutions. L-Encoder, on the other hand, is a shallow network that generates features at a high resolution (*e.g.* 1/4 of the original size).

In this module, we firstly pool the low-level source features $\mathbf{F}_{\mathrm{low}}^A$ to a resolution of 1/8 and perform a matrix product with the target features $\mathbf{F}_{\mathrm{low}}^B$ to generate the low-level correlation matrix $\mathbf{C}_{\mathrm{low}}$ with shape $(\frac{H}{8}, \frac{W}{8}, \frac{H}{4}, \frac{H}{4})$. Next, we utilize correlation lookup operation proposed in (Teed & Deng, 2020) to crop a local correlations $\mathbf{C}_{\mathrm{local}} \in \mathbb{R}^{\frac{H}{8} \times \frac{W}{8} \times (2r+1)^2}$ from the $\mathbf{C}_{\mathrm{low}}$ according to the initial flow map $\mathbf{M}$, whereas $r$ is the radius of neighbourhood. This process can be formulated as follows:

$$\mathbf{C}_{\mathrm{local}}^{\mathbf{P}} = \mathrm{CosineSim}(\mathbf{F}^{\mathbf{P}}, \mathbf{F}^{\mathcal{N}(\mathbf{P}+\Delta\mathbf{P}, r)}), \tag{8}$$

where $\mathbf{p} \in \mathbb{R}^{1 \times K \times 2}$ is the coordinates of the source keypoint pixels, $\Delta\mathbf{p} \in \mathbf{M}$ is the initial predicted flow of the pixel, and $\mathcal{N}(\cdot)$ indicates the local neighborhood of a pixel.

In this process, we specifically utilize local correlations instead of global correlations. This choice is made because long-range correlations are unnecessary for upsampling and can potentially have a negative impact on the results since $\mathbf{C}_{\text{low}}$ is generated from a shallow network and only contains local information. The local correlations contain information about the relationship between the source pixel and the surrounding pixels that could potentially correspond to it. The flow map and the local correlation are then upsampled to the full resolution by bilinear interpolation, and seperately encoded by a CNN head. Finally, they are concatenated and fed into a flow head to derive the final full-resolution flow map. The flow head is a small CNN and its architecture is shown in the appendix.

## 4.3 TRAINING OBJECTIVE

In Sec. 3.1 the simple baseline is trained with EPE loss. The model learns the pixel-level matching relationship between two images, aligning features of the same parts across different objects. However, in some cases, multiple parts within a single object may have similar semantics, such as multiple wheels of a car or multiple legs of an animal. This similarity can often confuse the model and lead to incorrect matching results. To address this challenge, we introduce triplet loss (Schroff et al., 2015) as an auxiliary loss to distinguish highly similar parts. Triplet loss is defined as

$$L(a, p, n) = \max(d(a, p) - d(a, n) + m, 0), \tag{9}$$

where $a, p, n$ is the anchor, positive sample, negtive sample of a triplet, $d$ is the distance function (*e.g.* L2 distance) and $m$ is the margin. Triplet loss and its variant have been widely adopted in face recognition (Schroff et al., 2015; Boutros et al., 2022), classification (Sohn, 2016) and re-identification (Cheng et al., 2016; Luo et al., 2020; Hermans et al., 2017). When constructing triplets, these methods typically use the overall embedding of the entire object as a sample. In contrast, we employ the features at the locations of the keypoints as samples. Let $\mathbf{p} = \{p_1, p_2, ..., p_N\}$ and $\mathbf{q} = \{q_1, q_2, ..., q_N\}$ denote coordinates of the source keypoints and target keypoints, the auxiliary loss can be defined as

$$\mathcal{L}_{Aux} = \frac{1}{N(N-1)} \sum_{i}^{N} \sum_{j \neq i}^{N} L(\mathbf{F}^{p_i}, \mathbf{F}^{q_i}, \mathbf{F}^{q_j}). \tag{10}$$

During training, we use loss weights $\lambda_1$ and $\lambda_2$ to balance two losses, fomulated as

$$\mathcal{L} = \lambda_1 \mathcal{L}_{EPE} + \lambda_2 \mathcal{L}_{Aux}. \tag{11}$$

## 5 EXPERIMENTS

### 5.1 DATASETS AND EVALUATION METRIC

**Datasets.** We conducted experiments on three standard benchmarks for semantic correspondence: PF-PASCAL, PF-WILLOW (Ham et al., 2016) and SPair-71k (Min et al., 2019). The PF-PASCAL dataset contains 2,941 / 308 / 299 image pairs for `train` / `val`/ `test` set respectively. The PF-WILLOW dataset exclusively consists of 900 image pairs for `test` split. Image pairs in both the PF-PASCAL and PF-WILLOW datasets exhibit minor viewpoint and scale variations, making them relatively straightforward for analysis. The SPair-71k dataset is a dataset with larger scale, constructed from 1,800 images spanning 18 categories from PASCAL VOC. It comprises totally 70,958 image pairs with visual annotations, including keypoints and their correspondence, bounding boxes, segmentation masks, and more. The `train` / `val` / `test` set contains 53,340 / 5,384 / 12,234 image pairs respectively. Due to its complex scenes, the SPair-71k dataset presents a greater challenge compared to the previous two datasets.

**Evaluation Metric.** Percentage of correct keypoints (PCK) is used to evaluate the models. Given a source image $\mathbf{I}^A$ and a target image $\mathbf{I}^B$ with their keypoint annotations $\mathbf{K}^A$ and $\mathbf{K}^B$. The model takes $\{\mathbf{I}^A, \mathbf{I}^B, \mathbf{K}^A\}$ as input and output prediction keypoints $\hat{\mathbf{K}}^B$. Then PCK is calculated by

$$\text{PCK}(\mathbf{I}^A, \mathbf{I}^B) @ \alpha_\tau = \frac{1}{M} \sum_{m=1}^{M} [\|\mathbf{K}^B - \hat{\mathbf{K}}^B\| \leq \alpha \cdot \theta_\tau], \tag{12}$$

| Sup. | Method | Backbone | Res. | SPair-71k PCK@$\alpha_{bbox}$ | | | PF-PASCAL PCK@$\alpha_{img}$ | | | PF-WILLOW PCK@$\alpha_{bbox-kp}$ | | |
|---|---|---|---|---|---|---|---|---|---|---|---|---|
| | | | | 0.01 | 0.05 | 0.10 | 0.05 | 0.10 | 0.15 | 0.05 | 0.10 | 0.15 |
| U | DINOv2 (Oquab et al., 2023) | DINOv2 | 224 | 1.3 | 20.2 | 40.6 | 47.6 | 76.5 | 85.9 | 28.8 | 61.2 | 79.6 |
| | DIFT (Tang et al., 2023) | SD | - | - | - | 52.9 | - | - | - | - | - | - |
| | SD-DINO (Zhang et al., 2023) | SD&DINOv2 | 960 | - | - | 62.9 | 72.1 | 86.0 | 90.6 | - | - | - |
| W | SFNet (Lee et al., 2019) | ResNet-101 | ori. | - | 26.2 | 50.0 | 78.6 | 91.7 | 95.3 | 43.0 | 70.9 | 83.9 |
| | NCNet (Rocco et al., 2018) | ResNet-101 | ori. | - | 29.1 | 50.7 | 78.7 | 92.9 | 96.0 | 43.2 | 72.5 | 85.9 |
| S | DHPF (Min et al., 2020) | ResNet-101 | 240 | 1.7 | 20.7 | 37.3 | 75.7 | 90.7 | 95.0 | - | 71.0 | - |
| | CHM (Min & Cho, 2021) | ResNet-101 | 240 | 2.3 | - | 46.3 | 80.1 | 91.6 | - | - | 69.6 | - |
| | CATs (Cho et al., 2021) | ResNet-101 | 256 | 1.9 | 27.9 | 49.9 | 75.4 | 92.6 | 96.4 | 40.7 | 69.0 | - |
| | CATs++ (Cho et al., 2022) | ResNet-101 | 512 | 4.3 | 40.7 | 59.8 | 84.9 | 93.8 | 96.8 | 47.0 | 72.6 | - |
| | TransforM. (Seung Wook Kim, 2022) | ResNet-101 | 240 | - | - | 53.7 | 80.8 | 91.8 | - | - | 65.3 | - |
| | D.Hyperfeat. (Luo et al., 2023) | SD | 512 | - | - | 64.6 | - | - | - | - | - | - |
| | ACTR (Sun et al., 2023) | iBOT | 256 | 4.3 | 42.0 | 62.1 | 81.2 | 94.0 | 97.0 | 42.7 | 69.9 | 84.1 |
| | ACTR$_h$ (Sun et al., 2023) | iBOT | 512 | - | - | 65.4 | 82.0 | 93.5 | 96.7 | - | - | - |
| | GeoAware-SC (Zhang et al., 2024) | SD&DINOv2 | 960 | 21.7 | 72.8 | 83.2 | 85.3 | 95.0 | 97.4 | - | - | - |
| | GeoAware-SC* (Zhang et al., 2024) | SD&DINOv2 | 960 | **22.0** | 75.3 | 85.6 | 85.9 | 95.7 | **98.0** | - | - | - |
| | Baseline | DINOv2 | 224 | 7.0 | 58.6 | 78.2 | 85.6 | 95.6 | 97.5 | 46.1 | 74.0 | 86.8 |
| | ViTSC | DINOv2 | 224 | 9.3 | 63.3 | 81.8 | **87.4** | **96.3** | 97.8 | **49.7** | **76.7** | **88.4** |
| | ViTSC | DINOv2 | 448 | 19.6 | **76.0** | **86.6** | 86.0 | 95.3 | 97.6 | 47.2 | 72.6 | 84.9 |

Table 3: Comparison with state-of-the-art methods on SPair-71k, PF-PASCAL and PF-WILLOW. The best results are in bold. The main results of our method are marked in purple . U, W and S in the first column mean unsupervised, weakly supervised and strongly supervised methods respectively. * means extra training data is used. Results on SPair-71k are trained on SPair-71k itself, while results on PF-PASCAL and PF-WILLOW are trained on PF-PASCAL.

where $M$ represents the number of keypoints, $\alpha$ is the ratio ranging from 0 to 1, $\theta_\tau$ is the base threshold and $[\cdot]$ indicates the Iverson bracket. $\theta_\tau$ is defined as $\theta_\tau = max(w_\tau, h_\tau)$ where $\tau \in \{img, bbox, bbox\text{-}kp\}$, indicating image, bounding box and minimum bounding box of keypoints respectively. Following previous convention, we use $\alpha_{bbox}$ for SPair-71k, $\alpha_{img}$ for PF-PASCAL and $\alpha_{bbox-kp}$ for PF-WILLOW. All the reported results are evaluated on the test set of the respective dataset.

## 5.2 IMPLEMENTATION DETAILS

We adopt pre-trained DINOv2-B/14 as the H-Encoder and pre-trained ResNet-18 (He et al., 2016) as the L-Encoder. DINOv2 generates high-level features at 1/14 size of the original image. `Layer4` in ResNet-18 is removed and stride of the convolution layers in `layer2` and `layer3` is set to be 1, so ResNet-18 generates low-level features at 1/4 size of the original image. We finetune the last 4 layers of DINOv2 and MAE, and the last 8 layers for iBOT and CLIP. We train all our models with at a resolution of 224×224 and evaluate them at a resolution of 224×224 and 448×448. We set loss weights $\lambda_1 = 1.0$ and $\lambda_2 = 10.0$ and the margin of triplet loss $m = 0.3$ We use Adam optimizer and the learning rate is set to 3e-6 for the backbones and 3e-5 for other modules. Our models are trained for 10 epochs when fine-tuned on SPair-71k and 50 epochs on PF-PASCAL. For ablation studies, all our models are trained for 10 epochs on SPair-71k and evaluated with a resolution of 224×224. All experiments are performed with a batch size of 16 on 4 RTX 4090 GPUs.

## 5.3 RESULTS

We present the quantitative results of our model in comparison with other methods on three standard benchmarks: SPair-71k, PF-PASCAL and PF-WILLOW in Tab. 3 and some qualitative results in the Fig. 2. To provide a more transparent comparison, we present details about the backbone of the model and the image resolution used during inference for each method. Our method achieves state-of-the-art performance on most metrics of SPair-71k, PF-PASCAL and PF-WILLOW. Our method surpasses previous methods in most metrics. Compared to the baseline, our method further gains a improvement of 2.3 / 4.7 / 3.6 PCK at threshold $\alpha = 0.01/0.05/0.10$. By considering Tab. 2, when we replace the backbone of previous methods with DINOv2 for a fair comparison, our model still achieves the best performance on SPair-71k. On PF-PASCAL, our method surpasses other methods with 87.4 / 96.3 / 97.8 PCK at threshold $\alpha = 0.05/0.10/0.15$.

| Enhancement | PCK@$\alpha_{bbox}$ | | |
|---|---|---|---|
| Method | 0.01 | 0.05 | 0.10 |
| None | 7.7 | 59.7 | 79.9 |
| SymmetricXu et al. (2022) | 8.0 | 61.4 | 80.6 |
| AsymmetricSun et al. (2023) | 8.4 | 61.8 | 80.9 |
| Interleaved | **9.3** | **63.3** | **81.8** |

Table 4: Ablation on different feature enhancement methods.

| #Layers | PCK@$\alpha_{bbox}$ | | |
|---|---|---|---|
| | 0.01 | 0.05 | 0.10 |
| 0 | 7.7 | 59.7 | 79.9 |
| 1 | **9.3** | **63.3** | **81.8** |
| 2 | 9.2 | 63.4 | 81.7 |
| 3 | 8.9 | 62.8 | 80.7 |

Table 5: Ablation on the number of feature enhancement layers.

| Upsampling | PCK@$\alpha_{bbox}$ | | |
|---|---|---|---|
| | 0.01 | 0.05 | 0.10 |
| Bilinear | 7.5 | 60.9 | 80.5 |
| No guidance | 8.6 | 62.5 | 80.8 |
| $C_{high}$ guided | 8.3 | 62.8 | 81.1 |
| $C_{low}$ guided | **9.3** | **63.3** | **81.8** |

Table 6: Ablation on the design of flow upsampling module.

| $\mathcal{L}_{EPE}$ | $\mathcal{L}_{Aux}$ | PCK@$\alpha_{bbox}$ | | |
|---|---|---|---|---|
| | | 0.01 | 0.05 | 0.10 |
| ✓ | | 9.0 | 63.0 | 81.0 |
| ✓ | ✓ | **9.3** | **63.3** | **81.8** |

Table 7: Ablation on the auxiliary loss.

| Method | Backbones | Res. | PCK@$\alpha_{bbox}$ | | |
|---|---|---|---|---|---|
| | | | 0.01 | 0.05 | 0.10 |
| ACTR | iBOT | 256 | 4.3 | 42.0 | 62.1 |
| ViTSC | iBOT | 224 | 5.1 | 44.0 | 63.2 |
| ViTSC | MAE | 224 | 6.1 | 47.8 | 66.7 |
| ViTSC | CLIP | 224 | 5.3 | 44.0 | 62.6 |
| ViTSC | DINOv2 | 224 | **9.0** | **63.0** | **81.0** |

Table 8: Impact of different backbones on our method.

To assess the generalization capabilities of the models, we evaluate their performance on PF-WILLOW using the model pre-trained on PF-PASCAL. The results demonstrate that our method not only achieves high performance on the dataset it was trained on, but also exhibits excellent generalization capabilities. Our model achieves 49.7 / 76.7 / 88.4 PCK at threshold $\alpha = 0.05/0.10/0.15$ respectively.

The resolution of the input images has a significant impact on the accuracy on SPair-71k, especially when $\alpha$ is very small, $e.g$ 0.05 or even 0.01. On SPair-71k, increasing the resolution from 224 to 448 leads to notable improvements on PCK, with increases of 10.3 / 12.7 / 4.8 at threshold $\alpha = 0.01/0.05/0.10$ respectively. However, evaluating at a higher resolution does not yield improvements on PF-PASCAL and PF-WILLOW. In fact it slightly downgrades the performance. We speculate that this discrepancy could be attributed to the relatively small size of the PF-PASCAL dataset. When fine-tuning the model with a small resolution on this dataset, it becomes more prone to overfitting due to the limited amount of available training data.

Compared to the previous state-of-the-art method GeoAware-SC, we surpass it on the vast majority of metrics. Notably, we only use a single pre-trained visual foundation model, a smaller resolution, and no additional training data.

## 5.4 ABLATION STUDIES AND ANALYSES

We conducted ablation experiments on various feature enhancement methods for the cross-perception module, the number of layers of the cross-perception module and the upsampling method to verify the effectiveness of our design. In ablations, all models are trained for 10 epochs and evaluated on SPair-71k at a resolution of 224, using a ViT-B as their backbone.

**Feature Enhancement Method.** The evaluation results for different features enhancement modules are presented in Tab. 4. The symmetric module, asymmetric module and interleaved module exhibit improvements of 0.7, 1.0 and 1.9 PCK@0.10 respectively. Additionally, we investigated the impact of stacking feature enhancement layers. The results in Tab. 5 indicate that one layer is the most suitable choice. Since the features extracted by the fine-tuned backbone are already of high quality, adding a single interleaved feature enhancement layer allows the source features and the target fea-

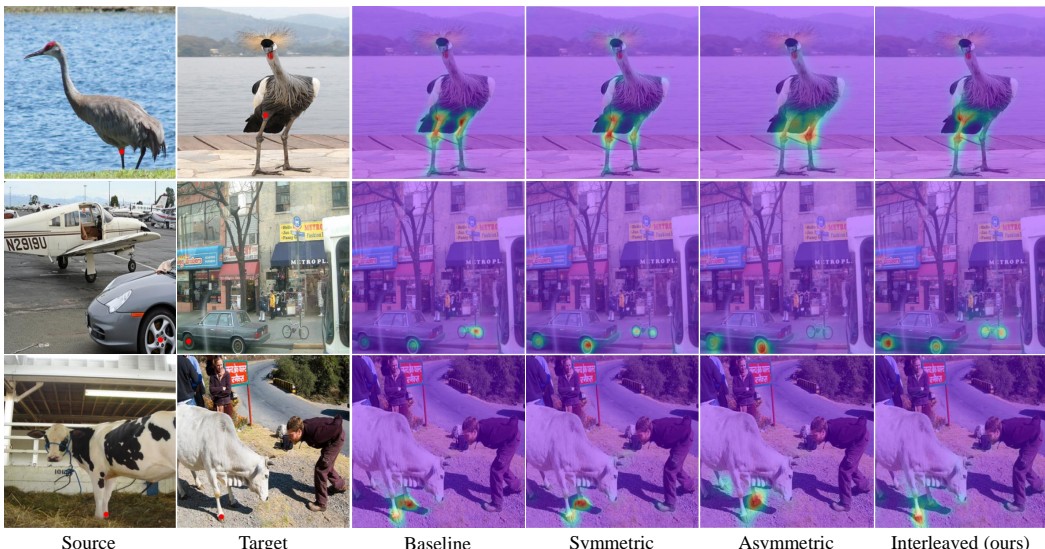

|        |        |          |           |            |                   |
| :----: | :----: | :------: | :-------: | :--------: | :---------------: |
| Source | Target | Baseline | Symmetric | Asymmetric | Interleaved (ours) |

Figure 3: The heatmap shows the correlation between the pixel in the source image and all pixels in the target image. The visualization proves that our interleaved feature enhancement module can eliminate ambiguity to a certain extent.

tures to gain awareness of each other's characteristics. However, incorporating more stacked layers may lead to the destruction of the original features from the backbone. We present visualizations of correlations in Figure 3, showcasing the effectiveness of our approach. Specifically, in scenarios where two objects or two parts of a single object exhibit high similarity within an image (*e.g.*two wheels on a car), our interleaved attention module successfully learns the appropriate correlations.

**Flow Upsampling Method.** Tab. 6 presents the impact of different upsampling methods. *No guidance* refers to employing a CNN directly on the bilinearly upsampled flow map to obtain the flow map at the full resolution. $\mathbf{C}_{high}$ *guided* utilizes the high-level features to calculate the correlation map used in the correlation lookup operation (as described in Sec. 4.2). $\mathbf{C}_{low}$ *guided* refers to our method, which employs low-level features correlation map to guide upsampling. The results indicate that *no guidance* already yields a slight increase in accuracy (+0.3 PCK@0.10), and our method further improves upon it (+1.3 PCK@0.10). However, $\mathbf{C}_{high}$ *guided* does not bring significant improvement compared with *no guidance* due to the lack of high-resolution information.

**Auxiliary Loss.** We test the impact of incorporating the auxiliary loss on the model's performance. The results show that introducing the auxiliary loss leads to an improvement of 0.8 PCK@0.10.

**Pre-trained Backbones.** To validate the compatibility of our matching module with other backbones, we train our model using various pre-trained ViTs as presented in Tab. 8. It is evident that different pre-trained backbones have a significant impact on the fine-tuning results on SPair-71k. Using iBOT as the backbone (same with ACTR), our model surpasses ACTR (63.2 vs 62.1 PCK@0.10) with a lighter matching module (1 layer vs 6 layers) and a lower resolution (224 vs 256).

## 6    CONCLUSION

In this work, we extensively investigate the utilization of pre-trained ViTs for semantic correspondence tasks. We construct a straightforward yet robust baseline that serves as an intuitive way to evaluate the performance of different pre-trained models on semantic matching tasks. Additionally, we introduce a novel model named ViTSC to further unleash the strength of pre-trained models. Through comprehensive experiments and visualization, we provide substantial evidence to demonstrate the effectiveness of our designs.

**Limitations.** Our method has a limitation in handling query pairs with very large pose or view discrepancies . To mitigate this, we will look into self-supervised and data augmentation methods, which can help enhance the robustness in these cases.

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

# A ADDITIONAL RESULTS AND ANALYSES

**Category-wise evaluation results.** We show the category-wise evaluation results of different methods on SPair-71kMin et al. (2019) at $\alpha = 0.10$ in Tab. 9. Our ViTSC achieves the best performance in most of the categories and outperforms the baseline in all categories. ViTSC demonstrates superior performance compared to previous state-of-the-art methods, particularly in categories like horse, motorbike, person, pottedplant, etc.

| Method | aero | bike | bird | boat | bottle | bus | car | cat | chair | cow | dog | horse | mbike | person | plant | sheep | train | tv | all |
|---|---|---|---|---|---|---|---|---|---|---|---|---|---|---|---|---|---|---|---|
| TransforM.Seung Wook Kim (2022) | 59.2 | 39.3 | 73.0 | 41.2 | 52.5 | 66.3 | 55.4 | 67.1 | 26.1 | 67.1 | 56.6 | 53.2 | 45.0 | 39.9 | 42.1 | 35.3 | 75.2 | 68.6 | 53.7 |
| CATs++Cho et al. (2022) | 60.6 | 46.9 | 82.5 | 41.6 | 56.8 | 65.1 | 50.4 | 72.8 | 29.2 | 75.8 | 65.4 | 62.5 | 50.9 | 56.1 | 54.8 | 48.3 | 80.8 | 74.9 | 59.8 |
| ACTRSun et al. (2023) | 65.0 | 48.5 | 82.3 | 50.4 | 55.9 | 65.3 | 63.1 | 72.8 | 35.8 | 74.1 | 70.3 | 68.9 | 58.6 | 57.1 | 46.8 | 49.5 | 84.4 | 73.3 | 62.1 |
| TransforM.[‡] | 83.1 | 67.9 | 87.4 | 66.1 | 71.1 | 86.8 | 85.1 | 88.1 | 67.5 | 85.0 | 83.1 | 77.8 | 72.6 | 75.2 | 71.0 | 67.8 | 88.9 | 89.9 | 78.3 |
| CATs++[‡] | 77.9 | 60.6 | 84.5 | 61.0 | 67.5 | 83.8 | 76.5 | 87.2 | 69.8 | 83.7 | 78.3 | 75.2 | 66.8 | 75.1 | 65.5 | 67.8 | 87.4 | 85.5 | 74.9 |
| ACTR[‡] | 82.7 | 60.2 | 87.4 | 71.4 | 63.0 | 87.2 | 82.5 | 88.0 | 68.3 | 83.7 | 81.6 | 75.6 | 68.9 | 62.1 | 58.4 | 64.8 | 89.5 | 81.9 | 74.9 |
| Baseline | 82.6 | 67.3 | 87.9 | 65.1 | 71.1 | 87.8 | 83.0 | 87.7 | 67.9 | 84.7 | 83.5 | 77.2 | 71.1 | 76.0 | 72.2 | 68.7 | 91.4 | 90.2 | 78.2 |
| ViTSC | 86.1 | 69.7 | 89.9 | 70.3 | 75.7 | 87.4 | 87.6 | 89.5 | 74.8 | 88.4 | 86.6 | 82.0 | 75.9 | 80.4 | 74.2 | 73.2 | 93.6 | 92.2 | 81.8 |
| ViTSC$_h$ | 91.4 | 74.2 | 96.5 | 75.2 | 77.8 | 92.0 | 87.8 | 93.3 | 80.0 | 94.0 | 92.9 | 88.8 | 84.0 | 88.7 | 80.1 | 76.8 | 95.4 | 94.9 | 86.6 |

Table 9: Category-wise evaluation results on SPair-71k at $\alpha = 0.10$. The best results and the second best results are emphasized with **bold** and underline formatting respectively. All models in the second group are evaluated at a resolution of 224. ViTSC and ViTSC$_h$ share the pretrained weights and the only difference between them is the evaluation resolution (224 vs 448). [‡] indicates models reproduced by us with DINOv2-B as the backbone (same with ViTSC).

**Efficiency.** We present the number of parameters, inference resolutions, respective inference time and PCK in Tab. 10. The table demonstrates that the baseline model, consisting only of a backbone, achieves both fast inference time and good PCK. On the other hand, a heavy matching module like ACTR's is not necessary. Our ViTSC model achieves a significant performance increase with an acceptable time overhead. Additionally, increasing the inference resolution can introduce a substantial time overhead, although it leads to performance improvement.

The scale of the backbone is also a key factor affecting the inference time and PCK. We test models using DINOv2-S and DINOv2-L as the backbone, denoted as ViTSC$_s$ and ViTSC$_l$. They achieve 72.0 and 85.4 PCK, respectively, at $\alpha = 0.10$. ViTSC$_l$ demonstrates comparable performance to ViTSC$_h$ while requiring significantly less time. Therefore, scaling the backbone is a more efficient approach compared to increasing the resolution when aiming to improve performance.

**Visualization.** We provide qualitative results in Fig. 5 and Fig. 6, which Visually demonstrates the performance of our model.

# B ADDITIONAL DETAILS

**Settings for the preliminary experiments.** In the preliminary experiments in Sec.3, all models employ DINOv2-B as their backbone. The baseline and ACTR only utilize the last layer features outputted by DINOv2. The original versions of TransforMatcher and CATs++ enhance features with

| Method | Backbone | #Params (M) Matching module | Total | Resolution | Inference time (ms) | PCK@$\alpha_{bbox}$ 0.10 |
|---|---|---|---|---|---|---|
| TransforM.Seung Wook Kim (2022) | 87.0 | 0.9 | 87.9 | 240 | 13.3 | 53.7 |
| CATs++Cho et al. (2022) | 44.5 | 5.5 | 50.0 | 512 | 52.7 | 59.8 |
| ACTRSun et al. (2023) | 85.8 | 86.5 | 172.3 | 256 | 15.2 | 62.1 |
| TransforM.[‡] | 86.6 | 0.8 | 87.4 | 224 | 7.9 | 78.3 |
| CATs++[‡] | 86.6 | 9.0 | 95.5 | 224 | 40.5 | 59.8 |
| ACTR[‡] | 86.6 | 86.5 | 173.1 | 224 | 15.2 | 74.9 |
| Baseline | 86.6 | 0 | 86.6 | 224 | 5.4 | 78.2 |
| ViTSC | 89.4 | 3.9 | 93.3 | 224 | 8.7 | 81.8 |
| ViTSC$_h$ | 89.4 | 3.9 | 93.3 | 448 | 43.9 | 86.6 |
| ViTSC$_s$ | 24.9 | 3.9 | 28.8 | 224 | 6.2 | 72.0 |
| ViTSC$_l$ | 305.9 | 4.0 | 309.9 | 224 | 16.6 | 85.4 |

Table 10: Comparison of efficiency between ViTSC and other methods. Inference time is tested on a single NVIDIA RTX 4090 GPU. [‡] indicates models reproduced by us with DINOv2-B as the backbone (same with ViTSC).

the multi-layer features of ResNet-101, which we have adapted to the ViT architecture. Transfor-Matcher and CATs++ make use of all features outputted by DINOv2, from the 1st to the 12th layer. Due to the patch size of 14 in DINOv2, we opted not to use an image resolution of 256, instead, all models operate at an image resolution of 224. All models are trained on the SPair-71k dataset for 10 epochs.

**Flow head.** The flow head in Sec. 4.2 is a small CNN and its architecture is shown in Fig. 4.

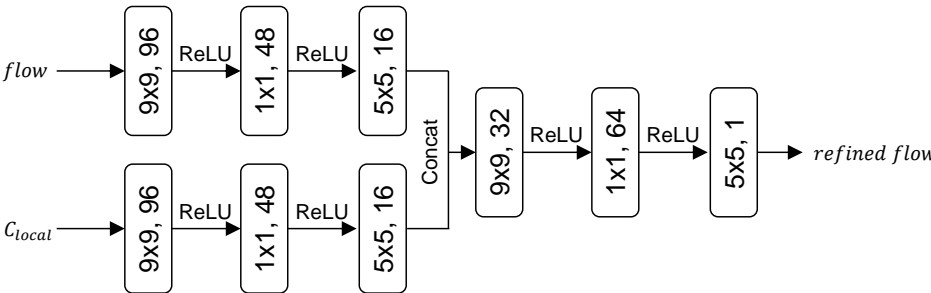

Figure 4: The architecture of the flow head.

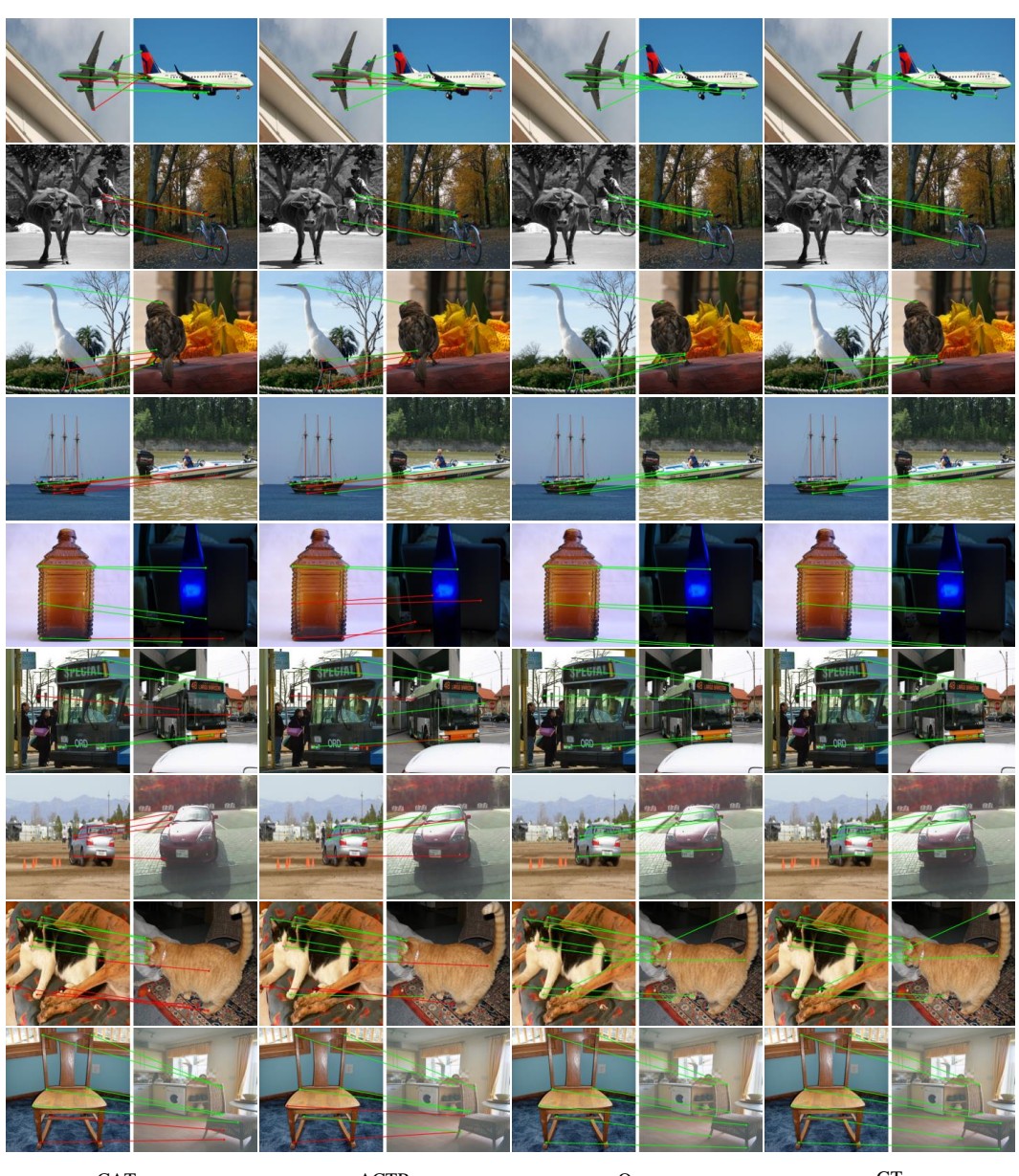

Figure 5: More visualization results on SPair-71k.

CATs++          ACTR          Ours          GT

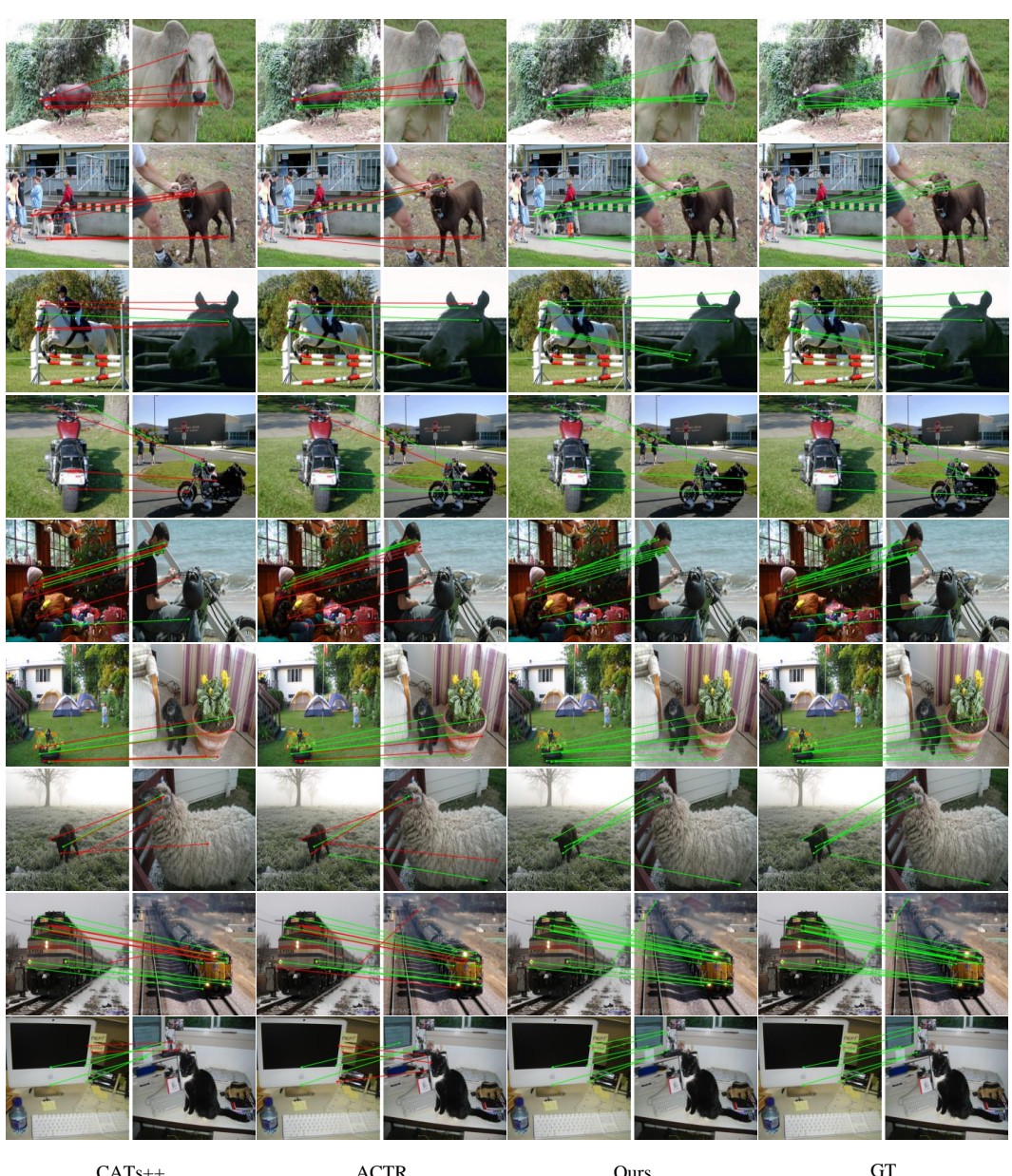

CATs++                    ACTR                    Ours                    GT

Figure 6: More visualization results on SPair-71k.

