# OpenReview forum: "Bridging the Gap between Semantic Correspondence and Robust Visual Representation"
_ICLR.cc/2025/Conference — Submitted to ICLR 2025_

### Official Review · Reviewer_hXia · 2024-10-28

**Soundness:** 3
**Presentation:** 3
**Contribution:** 2
**Rating:** 6
**Confidence:** 3

**Summary:**

This work proposed a two level representation learning method called ViTSC. In this framework, DINO V2 was used as  backbone followed with two branches namely high-level semantic encoder and low level texture encoder. For the high-level branch, correspondence was represented as semantic flow and was used to refine the correspondence for low-level branch. Semantic flow of both branches are summarized followed with the flow head. Results shown ViTSC performs will in generating reliable semantic correspondence. However, several analysis are required.

**Strengths:**

- The performance of ViTSC is attractive
- The paper is well organized

**Weaknesses:**

- Futher comparision of ViTSC and LoFTR is required. What is the main difference of ViTSC and LoFTR in pipeline design.
- Several Multi-Scale method for semantic correspondence such as MMNet and VAT have to be introduced in the related work. The difference of ViTSC for using high-level and low-level features compared with other multi-scale pipelines should be analyzed.
- For auxiliary loss, what is the main difference of designed loss compared with WarpC
- I have recognized the performance of ViTSC. However, what is the performance of previous SOTAs when using DINO V2, can better backbone contribute to all the semantic correspondence pipelines?

[1] LoFTR: Detector-free local feature matching with transformers

[2]  Cost aggregation with 4d convolutional swin transformer for few-shot segmentation

[3] Multi-scale matching networks for semantic correspondence

[4] Warp consistency for unsupervised learning of dense correspondences

**Questions:**

See the weakness part. The author should carefully discuss the design of pipeline and modules cpmpared with listed works.

---

### Official Review · Reviewer_hxN4 · 2024-10-31

**Soundness:** 1
**Presentation:** 1
**Contribution:** 1
**Rating:** 3
**Confidence:** 5

**Summary:**

This paper introduces a new framework called ViTSC, which aims to unleash the immense potential of self-supervised visual transformers for semantic correspondence. This task is crucial in computer vision, involving the prediction of semantic correspondences across images for different instances within the same category. ViTSC effectively enhances the model's matching capability on the data by leveraging the feature extraction abilities of self-supervised pretraining models. To address the limitations of existing semantic correspondence methods in feature quality and matching module design, ViTSC introduces three key components: a cross-attention module for aligning semantic features of the same parts in different images, an auxiliary loss module to distinguish similar objects, and a low-level correlation-guided upsampler for generating high-resolution flow maps. Systematic experiments demonstrate that these three components work together to enable ViTSC to perform effectively on the three standard benchmarks: SPair-71k, PF-PASCAL, and PF-WILLOW.

**Strengths:**

1.The paper compares ViTSC with several existing state-of-the-art methods, demonstrating performance improvements across multiple metrics, which enhances the persuasiveness of the research findings.
2.The proposed ViTSC framework is based on self-supervised pretraining models, which have a well-established theoretical foundation and can provide powerful feature representations. The paper provides a detailed description of the mathematical formulas and processes for the cross-attention module and correlation-guided upsampler, clearly illustrating how the algorithm works.

**Weaknesses:**

1. The paper only conducts experiments on three standard benchmarks: SPair-71k, PF-PASCAL, and PF-WILLOW, which may not cover a broader range of datasets, especially those that include more diverse scenes and conditions. This limits the comprehensive assessment of the model's generalization capabilities.
2. The paper does not adequately test the model's performance under extreme conditions, such as in complex scenarios involving extreme lighting, weather conditions, occlusion, or rapid motion, indicating a lack of scalability.
3. The experiments may not evaluate the model's real-time performance and computational efficiency in practical applications, which is particularly important for resource-constrained environments such as mobile devices.

4. Although the ViTSC framework improves upon existing methods, the paper lacks a detailed description of the innovation in the model architecture, especially in comparison with existing technologies. While the proposed model includes a cross-attention module and a correlation-guided upsampler, its novelty is insufficient, and the analysis of these modules' impact on the overall architecture is not in-depth.
5. The experimental section of the paper lacks specific analyses, such as an in-depth exploration of the reasons behind the model's good performance in comparative experiments.

**Questions:**

1. The paper only conducts experiments on three standard benchmarks: SPair-71k, PF-PASCAL, and PF-WILLOW, which may not cover a broader range of datasets, especially those that include more diverse scenes and conditions. This limits the comprehensive assessment of the model's generalization capabilities.
2. The paper does not adequately test the model's performance under extreme conditions, such as in complex scenarios involving extreme lighting, weather conditions, occlusion, or rapid motion, indicating a lack of scalability.
3. The experiments may not evaluate the model's real-time performance and computational efficiency in practical applications, which is particularly important for resource-constrained environments such as mobile devices.

4. Although the ViTSC framework improves upon existing methods, the paper lacks a detailed description of the innovation in the model architecture, especially in comparison with existing technologies. While the proposed model includes a cross-attention module and a correlation-guided upsampler, its novelty is insufficient, and the analysis of these modules' impact on the overall architecture is not in-depth.
5. The experimental section of the paper lacks specific analyses, such as an in-depth exploration of the reasons behind the model's good performance in comparative experiments.

---

> ### Comment · Reviewer_hxN4 · 2024-12-02
>
> I will keep my initial rating.

---

### Official Review · Reviewer_XuAS · 2024-11-01

**Soundness:** 3
**Presentation:** 2
**Contribution:** 2
**Rating:** 3
**Confidence:** 5

**Summary:**

This paper addresses the semantic correspondence problem by leveraging pre-trained vision transformers (DINOv2), referred to as ViTSC. ViTSC consists of  three key components: a cross-perception module, a correlation-guided upsampler, and an auxiliary loss based on triplet loss. The effectiveness of these simple yet effective modules is verified through comprehensive ablation studies.

**Strengths:**

* Good Motivation: Leveraging the pre-trained backbone is a strong motivation for enhancing performance on semantic correspondence benchmarks in practical applications. Table 1 verifies that the existing DINOv2 backbone demonstrates superior performance compared to other pre-trained models such as iBOT, MAE, and CLIP. Additionally, Table 2 illustrates that the simple baseline can outperform more complex methods that incorporate additional matching modules [Kim et al., CVPR 2022, Sun et al., CVPR 2023, Cho et al., PAMI 2022].
* Ablation Studies: The proposed modules are thoroughly verified on the baseline DINOv2 pre-trained backbone through ablation studies.
    1. Interleaved Attention Module: The interleaved attention module demonstrates superior effectiveness compared to symmetric and asymmetric baselines (Tables 4 and 5).
    2. Correlation-Guided Upsampler: The correlation-guided upsampler, guided by C_low, shows improved performance over simple bilinear upsampling and C_high guidance (Table 6).
    3. Auxiliary Triplet Loss: The auxiliary loss designed to distinguish keypoints in feature space via triplet loss proves to be effective (Table 7).

**Weaknesses:**

1. Lack of Novelty:
    * The interleaved attention module is a relatively simple modification of the cross-attention module input.
    * The correlation-guided upsampler has already been proposed in RAFT (Teed and Deng, ECCV 2020).
    * The auxiliary triplet loss is previously introduced in HardNet [A].

[A] Mishchuk et al., "Working hard to know your neighbor's margins," NeurIPS 2017.

Is there a difference in the way the modules proposed by RAFT or HardNet are applied or adapted in the semantic correspondence benchmarks covered in this paper?


2. Comparison to Stable Diffusion Backbones: Existing studies [B, C, D, E] suggest using stable diffusion backbones for solving semantic correspondence. The authors should include these baselines in Table 1 and provide a comparative discussion.
This can be experimented with in the standard public semantic correspondence benchmarks PF WILLOW, PASCAL, and SPair.

[B] Tang et al., "Emergent Correspondence from Image Diffusion," NeurIPS 2023.
[C] Hedlin et al., "Unsupervised Semantic Correspondence Using Stable Diffusion," NeurIPS 2023.
[D] Zhang et al., "A Tale of Two Features: Stable Diffusion Complements DINO for Zero-Shot Semantic Correspondence," NeurIPS 2023.
[E] Luo et al., "Diffusion Hyperfeatures: Searching Through Time and Space for Semantic Correspondence," NeurIPS 2023.

3. Limited Benchmark Testing: The ablation studies are primarily conducted on SPairs. Is there a reason why the ablation study was only performed on the SPair-71k? It is crucial to validate the results on other semantic correspondence benchmarks such as PF-WILLOW and PASCAL to ensure broader applicability.

4. Lack of Explanation for Figure 3: The manuscript does not explain Figure 3 adequately. It is essential to clarify why the interleaved feature enhancement is superior to symmetric and asymmetric feature enhancement from this perspective. In addition, the examples in Figure 3 visualization only covers left/right reflection symmetry, but in the real world, there exists rotational symmetry frequently. Does the proposed module  handle the rotational symmetry cases?

5. Broken paper formatting: Table 4 citation is broken. Table 3 TransforMatcher row is wrongly cited.

**Questions:**

Firstly, the proposed modules, such as the correlation-guided upsampler and the auxiliary loss, lack novelty, as they are incremental improvements over existing methods like RAFT and HardNet.

Secondly, the paper fails to compare its approach with recent advancements using stable diffusion backbones, which are shown to be effective in solving semantic correspondence problems. This omission leaves the evaluation incomplete.

Additionally, the ablation studies are primarily conducted on the SPair-71k dataset, raising concerns about the generalizability of the results to other benchmarks like PF-WILLOW and PASCAL.

To summarize, this paper lacks novelty as the proposed modules are incremental improvements over existing methods, while the paper leverages pre-trained vision transformers (DINOv2) for semantic correspondence. Additionally, the comparison with stable diffusion backbones is missing, and the limited benchmark testing on SPair-71k raises concerns about the generalizability of the results.

---

### Official Review · Reviewer_otsq · 2024-11-04

**Soundness:** 3
**Presentation:** 3
**Contribution:** 3
**Rating:** 5
**Confidence:** 4

**Summary:**

This paper introduces a simple yet effective framework named ViTSC to unlock the substantial potential of self-supervised vision transformers for semantic correspondence. The paper introduces three key components: a cross-perception module to align semantic features of the same part from different images while preserving the original representation as much as possible, an auxiliary loss to eliminate ambiguity from semantically similar objects, and a low-level correlation-guided upsampler to generate high-resolution flow maps for precise localization. Extensive experiments have demonstrated the effectiveness of the method.

**Strengths:**

1. The paper conducts extensive comparative experiments to demonstrate the effectiveness of the proposed method. The paper is skillfully written, well-articulated, and structurally clear.

2. The paper is quite innovative to some extent, the ablation studies are detailed and the results are analyzed in depth.

**Weaknesses:**

1. In the abstract section, the paper repeatedly emphasizes the complexity of the matching module design in existing methods and underscores that the proposed method is both simple and effective. However, the experiments lack an analysis of the matching module's complexity, limiting the validation of the paper’s claims.

2. I strongly recommend that the authors make the original code publicly available upon submission to enhance the credibility of the proposed method.

3. The introduction of this paper outlines three issues that need to be addressed; however, the visualization experiments lack comparative results demonstrating performance improvements in scenarios related to these three issues.

4. The naming of this high-resolution low-level correlation-guided upsampling module is not concise enough and seems a bit long.

**Questions:**

See above weaknesses.

---

> ### Comment · Reviewer_otsq · 2024-11-25
>
> After reading the author's response and other reviewers' comments, I will reduce my initial score.

---

### Meta-Review · Area_Chair_kZXE · 2024-12-21

**Metareview:**

This paper introduces a simple yet effective framework, ViTSC, aimed at unlocking the substantial potential of self-supervised vision transformers for semantic correspondence. The authors have not provided a rebuttal, and all reviewers have recommended rejection. After careful consideration, the Area Chair (AC) agrees with the reviewers' assessments and makes the decision to reject the paper.

**Additional Comments On Reviewer Discussion:**

A consensus has been reached by reviewers to reject this paper, please see the discussion section for more details.

---

### Decision · Program_Chairs · 2025-01-22

Reject